# Insights into the New Cancer Therapy through Redox Homeostasis and Metabolic Shifts

**DOI:** 10.3390/cancers12071822

**Published:** 2020-07-07

**Authors:** Dong-Hoon Hyun

**Affiliations:** Department of Life Science, Ewha Womans University, Seoul 03760, Korea; hyundh@ewha.ac.kr; Tel.: +82-2-3277-6635

**Keywords:** cancer, oxidative stress, glycolysis, oxidative phosphorylation, Nrf2-Keap1, NQO1

## Abstract

Modest levels of reactive oxygen species (ROS) are necessary for intracellular signaling, cell division, and enzyme activation. These ROS are later eliminated by the body’s antioxidant defense system. High amounts of ROS cause carcinogenesis by altering the signaling pathways associated with metabolism, proliferation, metastasis, and cell survival. Cancer cells exhibit enhanced ATP production and high ROS levels, which allow them to maintain elevated proliferation through metabolic reprograming. In order to prevent further ROS generation, cancer cells rely on more glycolysis to produce ATP and on the pentose phosphate pathway to provide NADPH. Pro-oxidant therapy can induce more ROS generation beyond the physiologic thresholds in cancer cells. Alternatively, antioxidant therapy can protect normal cells by activating cell survival signaling cascades, such as the nuclear factor erythroid 2-related factor 2 (Nrf2)-Kelch-like ECH-associated protein 1 (Keap1) pathway, in response to radio- and chemotherapeutic drugs. Nrf2 is a key regulator that protects cells from oxidative stress. Under normal conditions, Nrf2 is tightly bound to Keap1 and is ubiquitinated and degraded by the proteasome. However, under oxidative stress, or when treated with Nrf2 activators, Nrf2 is liberated from the Nrf2-Keap1 complex, translocated into the nucleus, and bound to the antioxidant response element in association with other factors. This cascade results in the expression of detoxifying enzymes, including NADH-quinone oxidoreductase 1 (NQO1) and heme oxygenase 1. NQO1 and cytochrome b5 reductase can neutralize ROS in the plasma membrane and induce a high NAD^+^/NADH ratio, which then activates SIRT1 and mitochondrial bioenergetics. NQO1 can also stabilize the tumor suppressor p53. Given their roles in cancer pathogenesis, redox homeostasis and the metabolic shift from glycolysis to oxidative phosphorylation (through activation of Nrf2 and NQO1) seem to be good targets for cancer therapy. Therefore, Nrf2 modulation and NQO1 stimulation could be important therapeutic targets for cancer prevention and treatment.

## 1. Introduction

Cancer cells exhibit typical biological characteristics that result from genetic mutations and altered regulatory systems that transform normal cells into cancer cells [1]. These transformed cells have a different microenvironment than do normal cells, including a high ATP demand (to proliferate) and low O_2_ supply due limited generation of new blood vessels. To support these changes, cancer cells must induce metabolic reprogramming from oxidative phosphorylation to glycolysis [2,3]. This change can be induced by activating oncogenes such as Ras, and inhibiting tumor suppressor genes such as p53 [4,5]. Although many current cancer therapies are based on glycolysis inhibition, these approaches can subsequently impair mitochondrial function.

The electron transport chain (ETC), containing complexes I, II, III, and IV, which comprise the main part of oxidative phosphorylation, plays a crucial role in cancer cell proliferation, survival, and metastasis because complex I exhibits pro-tumorigenic functions [6]. Recent studies have investigated the use of mitochondrial complex I-targeting drugs such as biguanides and metformin. In cancer cells with mutated mitochondrial DNA (mtDNA), the mitochondrial complex I is affected by biguanides [7]. One group found that a combined treatment with another mitochondria targeting drug, mito-carboxy-proxyl (Mito-CP), and the glycolysis inhibitor 2-deoxyglucose (2-DG) synergistically induced cancer cell death [8]. Therefore, it is important to identify other medications that specifically target glycolysis or oxidative phosphorylation in cancer treatment.

This review focuses on molecular mechanisms in the relationships among increased ROS, altered intracellular signaling, and modified energy metabolism in cancer cells and their implications for new cancer therapy strategies.

## 2. Oxidative Stress and the Antioxidant Defense System

Eukaryotic cells generate ATP mainly through aerobic respiration in the mitochondria, which produce several compounds including reduced nicotinamide adenine dinucleotide (NADH), reduced flavin adenine dinucleotide (FADH_2_), and other intermediates from the citric acid cycle [9]. Most of these compounds are beneficial to cells. However, less than 5% of them are reactive species (RS) that can be harmful to cells if their levels are elevated [10]. Low levels of reactive species (which are converted from O_2_ during oxidative phosphorylation) are required for normal cellular physiology, including signal transduction, enzyme activation, gene expression, and post-translational modification. Oxidative stress is an imbalance between the production of reactive species and the antioxidant defense system in cells, which can lead to biomolecule damage.

RS are produced both inside and outside of cells. Several potential external sources of oxidative stress include physical radiation (e.g., X-rays and ultraviolet), chemical compounds (e.g., transition metals, smoking, and pollutants), and high-intensity exercises. Intracellular sources of oxidative stress are enzymes responsible for electron transport, hypoxia, tumor necrosis factor α (TNF- α), and other growth factors. In fact, the mitochondria are the main source of oxidative stress in cells because of the way of that the ETC is linked to ATP production. RS are divided into the following four categories: reactive oxygen species (ROS), reactive nitrogen species (RNS), reactive chlorine species, and reactive bromine species [11]. These RS can be very toxic because of their high reactivity and short half-life. ROS, such as superoxide anion, hydrogen peroxide, and hydroxyl radical, are produced during mitochondrial oxidative phosphorylation, which uses O_2_ as an electron acceptor [12]. RNS, including nitric oxide (NO^•^), are also generated through intracellular metabolism such as nitric oxide synthase (NOS) and can be converted to peroxynitrite by hydroxyl radical [13].

The toxic effects of ROS on cells are usually determined by their antioxidant capacity. Oxidative stress occurs when the pro-oxidant/antioxidant balance is broken, which leads to damage to intracellular macromolecules such as DNA, lipids and proteins [14]. 8-Hydroxy-2′-deoxyguanosine (a marker of DNA oxidation) induces DNA mutation and then causes aging and carcinogenesis [15,16]. 4-Hydroxy-2,3-nonenal and 8-isoprostane (end products of lipid peroxidation) can attack the cell membrane with enriched unsaturated fatty acids and cause further production of lipid peroxides, which increases membrane permeability and cell death [17,18]. Protein carbonyls (a marker of protein oxidation) and 3-nitrotyrosine (a marker of protein nitration) are also generated by ROS and can cause impaired 3-dimensional protein structures and consequent loss of protein function [19,20,21].

The antioxidant defense system scavenges a variety of RS in cells using endogenous (non-enzymatic and enzymatic) and exogenous (mainly non-enzymatic) pathways (Figure 1) [14]. Glutathione (GSH), α-lipoate, coenzyme Q (CoQ), urate, and melatonin are non-enzymatic endogenous antioxidant agents. Superoxide dismutase (SOD), catalase, glutathione peroxidase, glutathione reductase, and heme oxygenase (HO) are enzymatic endogenous antioxidant system. Although the intracellular antioxidant system is well balanced with oxidative input, its antioxidant capacity attenuates with age. The imbalance between oxidants and antioxidants can also result in necrosis, not only apoptosis. Flavonoids, catechins, and polyphenols are plant-based exogenous antioxidant molecules, which usually come from food and beverages because human cells have no biosynthetic machinery for them [22].

Oxidative stress is a part of the pathological alterations seen in neurodegenerative diseases, such as Alzheimer’s disease (AD), Parkinson’s disease (PD), and amyotrophic lateral sclerosis (ALS); cardiovascular diseases (e.g., cardiomyopathy, heart failure, and muscular atrophy); metabolic syndromes (e.g., hypertension and hypercholesterolemia); aging; and cancer [23,24]. Under normal conditions, cells produce very low levels of ROS. During acute inflammation, ROS released from swollen or injured cells adversely affect normal cell physiology. In states of chronic inflammation, very high ROS levels are maintained for a long time, which alter biomolecules. The mitochondria are particularly vulnerable to oxidative stress because they are the major sites of ROS production. In addition, mtDNA is less tightly packed than nuclear DNA, and mtDNA repair systems and mitochondrial antioxidant capacity are lower than those in the cytosol [25]. In fact, mitochondrial dysfunction appears in the early stages of many diseases, and tends to be associated with decreased antioxidant defenses and increased oxidative damage [26,27,28,29]. An accumulation of mutated mtDNA can impair the mitochondrial complexes [30,31], and thereby cause dysfunction of the mitochondrial complex I activity in AD, PD, ALS, and cancer [32,33,34]; defective activities of complex II and III in ALS [35]; and reduced complex III activity in aged hearts [36]. Mitochondrial dysfunction can also cause further oxidative stress and ATP shortages, which secondarily affect other biochemical pathways [37,38].

## 3. Cancer Microenvironment and Survival Mechanisms

Cancer cells are not isolated in the body but are surrounded by different types of cells. The cancer microenvironment (CME) consists of the blood vessels, neighboring cells that contribute to signaling and immunity, and extracellular matrix (ECM) surrounding cancer cells [39,40]. Cancer cells secrete signaling molecules and interact with the CME, leading to angiogenesis and immune tolerance. Carcinoma-associated fibroblasts (CAFs) are heterogeneous groups and derived from normal fibroblasts, smooth muscle cells, epithelial mesenchymal transition (EMT), and endothelial mesenchymal transition. CAF function is strictly controlled toward tumorigenesis by cancer cells [41,42].

Angiogenesis, which is the process of forming new blood vessels from existing ones, plays an important role in cancer cell proliferation and metastasis [43]. Cancer cells and CAFs produce signaling molecules involved in angiogenesis such as angiopoietin-1, vascular endothelial growth factor (VEGF) and its receptor, fibroblast growth factor, platelet-derived growth factor, and matrix metalloproteinase [44,45]. Levels of VEGF, a key molecule in angiogenesis, can also be increased by hypoxia and ROS production [46,47]. NADPH oxidase 1 (NOX1), which is overexpressed in colon and prostate cancers, increases levels of VEGF, VEGF receptor, and matrix metalloproteinase 3 (MMP-3) [48]. MMP-3 is involved in ECM breakdown [49].

Usually, cancer cell proliferation is suppressed by the immune system. Cancer cells can grow only when the immune function become dysfunctional (e.g., by attenuated antigen presentation, impaired immune cell trafficking, chronic inflammation, or metabolic restrictions that inhibit immune cells) [50]. Transforming growth factor β secreted by CAFs binds to EMT, induces metastasis, and inhibits cytotoxic T cells and natural killer cells [51,52]. Modest levels of ROS can cause inflammation, which promotes the initiation and development of cancer [53]. Resting T cell metabolism depends on aerobic oxidative phosphorylation. However, the metabolism of effector T lymphocytes relies on more aerobic glycolysis, resulting in competition between effector T cells and cancer cells [54]. p53, a tumor suppressor protein, is frequently mutated in cancer cells [55,56], and local activation of p53 in the CME can overcome immune suppression and enhance anti-cancer activity [57].

Alterations in the ECM components or their receptors are found in many cancers. The deposition of collagen types I, II, III, V, and IX is increased during tumorigenesis [58]. Heparan sulfate and CD44, which are responsible for growth factor signaling, are overproduced in cancer cells [59,60]. CD44 plays a key role in melanoma cell proliferation, which is induced by hyaluronic acid through the activation of mitogen-activated protein kinase (MAPK), Rac (a member of the Rho family of small GTPases), and phosphoinositol-3 kinase (PI3K) signaling pathways, which are essential for cancer cell survival [61]. Lysyl oxidase (LOX), which cross-links collagen fibers and other ECM components and increases tissue stiffness, is upregulated in many cancers including breast cancer and colorectal cancer [62,63]. Increased collagen cross-lining and ECM stiffness activates PI3K and extracellular-regulatory kinase (ERK) signaling pathways, and stimulates Neu-mediated oncogenic transformation [64]. Some studies suggest that LOX is essentially involved in cancer cell invasion and progression through ECM remodeling [65,66]. Therefore, the CME initiates tumorigenesis, promotes cancer cell proliferation, and induces metastasis by controlling angiogenesis, immune functions and the ECM components.

Extremely fast cancer cell proliferation has some restrictions: high ATP demand, shortage of O_2_ supply, and high ROS levels. In fact, increased ROS levels are involved in the development and progression of cancer cells by modifying energy metabolism through the following signal transduction pathways (Figure 2) [67]. First, cancer cell proliferation can be promoted through the activation of ERK1/2 and ligand-independent receptor tyrosine kinase (RTK) [68,69]. Second, cancer cells can evade apoptotic cell death through the stimulation of nuclear factor kappa-light-chain-enhancer of activated B cells (NF-κB) and PI3K/Akt [70,71]. Third, cancer cells release metalloproteinase into the ECM, after which they can invade the surrounding tissues. Fourth, cancer cells release VEGF and angiopoietin to make new blood vessels [72,73]. With regard to the prevention of cancer cell proliferation, it is important to maintain ROS levels below a certain point and keep mitochondrial function intact. The factors responsible for the antioxidant system are nuclear factor erythroid 2-related factor 2 (Nrf2), Kelch-like ECH-associated protein 1 (Keap1), Ras, p65, MAPK, and p38 [74,75,76]. The Nrf2-Keap1 pathway is one of the key regulators of the antioxidant response.

As described earlier, genetic factors contribute to the cellular transformations that lead to cancer development. Cancer cells modestly increase ROS levels; however, the ROS levels stay low enough that they do not damage the cancer cells. Cancer cells can also adapt to a new redox environment through mitochondrial reprogramming [77]. Cancer cells have more pathways to overcome oxidative damage than normal cells do because their high energy requirements stimulating oncogenes and attenuating tumor suppressor genes. The Ras pathway is the most important process associated with oxidative stress and cancer [78]. Mutations in Ras increase ROS levels, which causes DNA damage and transformation. These mutations occur in H-Ras, K-Ras, N-Ras, M-Ras, and R-Ras [79,80,81,82,83]. The NADPH oxidase system plays a crucial role in mutated Ras-induced oncogenicity, because knockdown of the system attenuates Ras^G12V^-induced DNA damage [84]. Human telomerase reverse transcriptase (hTERT), which is located in the mitochondria, is also involved in oxidative stress. When hTERT is overexpressed, the ROS levels are increased through recruitment of the mitochondria. Furthermore, inhibition of hTERT causes mitochondria-dependent apoptosis [85,86]. Inactivation of tumor suppressor genes (e.g., p53) can also increase ROS production. In fact, mutations in p53 are frequently identified in a variety of human cancers [87].

## 4. Metabolic Shifts in Cancer: Glycolysis and Oxidative Phosphorylation

Glycolysis is the common initial step in cellular energy metabolism. Pyruvate, which is generated from glucose during glycolysis, can enter one of two pathways depending on whether cells undergo fermentation (anaerobic) or mitochondrial respiration (aerobic). The Warburg effect is the phenomenon by which cancer cells can produce ATP by enhanced glycolytic activity even under high concentrations of O_2_ (20.9%) [88,89]. Recently, the molecular mechanism of the Warburg effect has been applied to cancer diagnosis. Specifically, positron-emission tomography using 2-[^18^F]fluoro-2-deoxy-D-glucose is used to detect areas with increased glycolysis in the body compared with controls [90]. Interestingly, this technique can also be used to assess decreased glucose utilization in the brain [91]. The Warburg effect, or high glucose demand, is interpreted as enhanced glycolysis and accompanied by increased activity of glycolytic enzymes.

There are a few explanations for why tumor cells mainly use glycolysis for metabolism, rather than mitochondrial respiration [77]. First, ATP production in cancer cells is less dependent on the mitochondria than it is in normal cells. This difference means that cancer cells already have high ROS levels so they need to prevent further ROS generation in the mitochondria [92,93]. Second, cancer cell proliferation occurs faster than angiogenesis in the surrounding environment; therefore, cancer cells must adapt to a shortage of O_2_ [94,95]. Third, high levels of glycolysis in cancer cells are connected to increased production of the substrate for the PPP [96,97]. The PPP, in turn, provides substrates for nucleotide biosynthesis during cancer cell division. It also generates NADPH, which prevents oxidative stress during reductive biosynthetic pathways.

Mitochondrial dysfunction is one of the remarkable characteristics identified in cancer cells. Alterations in mtDNA, such as point mutations and copy number changes, can compromise the mitochondrial ETC in cancer cells, and make the cells more resistant to anticancer agents [98,99]. These mitochondrial mutations can also make cancer cells more aggressive, overall. Mutations in mtDNA induce alterations in the mitochondrial ETC. This impairment causes tumorigenesis and increased ROS production, especially in mitochondrial complexes I and III [12,100]. These ROS promote cancer cell metastasis, which can be suppressed by ROS scavengers [101]. However, too much ROS can also damage cancer cells, so they need to attenuate mitochondrial functions so that they do not generate further ROS. The portion of ATP production that occurs through oxidative phosphorylation in cancer cells is approximately 80%, compared with 90% in normal cells [102,103]. Under hypoxic conditions, these levels diminish to around 30%. Increased ROS levels caused by defective mitochondrial ETC occur in cancer and age-related diseases, suggesting a similar relationship between mitochondrial dysfunction and oxidative stress in cancer and age-related diseases [37]. In addition, impairments in the citric acid cycle, such as abnormal succinate dehydrogenase, are linked to oncogenesis [104]. Silencing sirtuin 3, a mitochondrial deacetylase, can impair mitochondrial biogenesis and metabolism, causing cancer progression [105]. Thus, defective mitochondria can stimulate cytosolic signaling pathways and alter nuclear gene expression by changing the levels of ROS, [Ca^2+^]_i_ or onco-metabolites, leading to cancer cell transformation and progression.

The following signaling molecules play a role in regulating glycolysis and oxidative phosphorylation depending on energy demand and the microenvironment of the surrounding cells; hypoxia-inducing factor 1α (HIF-1α), von Hippel–Lindau tumor suppressor (VHL), p53, and cytoplasmic polyadenylation element binding (CPEB) protein. Cancer cells overcome the hypoxic conditions caused by their high proliferation and low angiogenesis rate through upregulating proangiogenic proteins. Under normoxic conditions, HIF-1α complexed with VHL binds to an E3 protein ligase, which results in polyubiquitination and proteasomal degradation [106]. Under hypoxic conditions, HIF-1α accumulates, dimerizes with HIF-1β, translocates into the nucleus, and induces the expression of target genes in the hypoxia response element (HRE) (Figure 3). In cancer cells, certain levels of ROS can induce NOSwhich then activates survival mechanisms through the PI3K/Akt and MEK/ERK pathways. HIF-1α can be phosphorylated through the MAPK signaling pathway. In addition, most solid cancers have hypoxic conditions in their cores. Therefore, cancer cells up-regulate other proangiogenic molecules (e.g., VEGF and erythropoietin) [72,73] and transcription factors that are responsible for glycolysis and cell survival [107,108].

Compensatory energy metabolism is required for the survival of cells lacking functional mitochondria, such as ρ^0^ cells [109], p53-deficient cells [110], and fibroblasts without CPEB protein [111]. The ρ^0^ cells are depleted of mtDNA and have enhanced glycolytic activity and a high NAD^+^/NADH ratio in association with an alternative pathway called the plasma membrane redox system (PMRS). The knockdown of p53 induces mtDNA deficiency and impaired redox homeostasis. Loss of CPEB can causes cell senescence, higher glycolysis and a lower number of the mitochondria and amount of p53. This altered energy metabolism is also seen in many cancer cells [112,113,114].

## 5. Redox Homeostasis as a Cancer Therapeutic Target

Many cancer cells have relatively high levels of ROS and adapt to that new microenvironment. This new balance can be broken by supplementary antioxidant therapy such as polyphenol [115]. However, it is important to add a supplementary antioxidant at a concentration appropriate to induce cellular stress adaptation responses, such as along the Nrf2-Keap1 pathway [116]. Antioxidant molecules at excessive concentrations can actually act as pro-oxidants, and then cause opposing effects in cells. Nrf2 activators may be a good candidate for cancer therapy, because these activators can reduce oxidative stress and inflammation. In addition, electrophilic modifiers of Keap1 and inhibitors of its interaction with NRF2 could also be novel therapeutic targets [117] Other ROS scavengers, including metal-base antioxidants, can move the redox balance to its original levels and thereby inhibit cancer cell growth [118]. Although these treatments could be effective, future investigations are needed regarding their target specificity, pharmacodynamics, efficacy, and safety.

Another approach to cancer therapy focuses on increasing ROS levels above the acceptable threshold in highly proliferating cancer cells [119]. Typically, cancer cells have higher levels of ROS than normal cells, which contribute to oncogenic pathways such as initiation, proliferation, and metastasis. However, ROS at toxic levels can cause too much oxidative damage, activate cancer cell death mechanisms, and enable drug resistance. For example, 5-fluorouracil and oxaliplatin are anticancer drugs that interfere with DNA replication and thus inhibit cell division. However, these medications are also associated with increased ROS production [120]. Similarly, the mitotic inhibitor Paclitaxel increases NADPH oxidase activity and then produces more ROS (e.g., O_2_^•−^ and H_2_O_2_), which ultimately slows cancer cell proliferation [121,122]. Bezielle (BZL101), which is isolated from *Scutellaria barbata*, produces a lot of ROS in cancer cells, which causes DNA damage and PARP cleavage and results in mitochondria-dependent apoptosis [123,124]. However, these treatments can affect both cancer cells and neighboring cells. Therefore, chemotherapeutic drugs are often combined with radiotherapy to increase their efficacy and decrease the side-effects on normal cells.

Phosphoglucose isomerase is the first target of glycolysis inhibition. 2-DG, a glycolysis blocker, competitively inhibits the production of glucose-6-phosphate from glucose (Figure 1). In one study, treatment with 2-DG dramatically decreased the proliferation and immortality of cancer cells [125]. Hexokinase can also be inhibited by ionidamide and 3-BP, which inhibits the growth of many cancer cells, including liver, gastric, and prostate cancers [126,127,128]. BZL101 not only inhibits glycolysis, but also oxidative phosphorylation [129]. The inhibition of glycolysis at the final step by pyruvate kinase is also a selective target. High ROS levels attenuate the activity of pyruvate kinase M2 (PKM2) and shift the metabolic pathway to the PPP to generate more NADPH and ribose. Therefore, a PKM2 activator can interfere with cancer cell metabolism. Although many reports have shown that anti-glycolytic drugs could be an effective therapy for cancer cells, these chemicals are cytotoxic against normal cells. In order to address this issue, mitochondrial functions must be considered. Combination therapy, using both a mitochondria-targeting drug and 2-DG, is more powerful than 2-DG alone. For example, Mito-Q and Mito-CP have a synergistic effect with 2-DG to induce breast cancer cell death [8].

## 6. Improvement of Mitochondrial Functions as a Cancer Therapeutic Target

The mitochondrial inner membrane contains the complexes responsible for the ETC that regulates oxidative phosphorylation and metabolite transport between the mitochondrial matrix and the cytosol. The targets of mitochondrial drugs include the mitochondrial complexes, mitochondrial permeability transition pores, and mtDNA. The molecular structure of the mitochondrial targeting drugs consists of a targeting moiety with the lipophilic part and a functional moiety (such as quinone, vitamin E, or tempol) [130]. One study based on the two-dimensional nature of bioenergetics in pancreatic cancer cells showed that the targeted inhibition of glycolysis and enhanced mitochondrial functions could effectively treat cancer cells [131].

Cells have compensatory mechanisms to survive despite a limited ATP supply. For example, cells can generate ATP even when energy is scarce (e.g., mitochondrial dysfunction and strenuous muscle activity) by stimulated glycolysis and fermentation. As a result, when cells receive supplemental pyruvate and uridine, they can survive without mitochondrial ATP production. This phenomenon is similar to the events that increase glycolysis and decrease oxidative phosphorylation in cancer cells to prevent further ROS production. These mechanisms suggest that inducing ROS levels beyond the threshold in cancer cells could cause cancer cell death (as long as the ROS levels are not enough to cause harm to the surrounding normal cells) [132,133]. Therefore, inducing a metabolic shift from the use of glycolysis to oxidative phosphorylation to produce ATP is a potential cancer therapy.

The NAD^+^/NADH ratio is an important regulator of the histone deacetylase of silent information regulator 2 (Sir2) in yeast and NAD-dependent deacetylase sirtuin-1 (SIRT1, a mammalian form of Sir2). A high NAD^+^/NADH ratio regulates mitochondrial respiration and cell life-spans [134,135]. Cells can regulate cellular redox homeostasis by maintaining a high NAD(P)^+^/NAD(P)H ratio by stimulating PMRS [136]. Alternatively, ρ^0^ cells generate lower levels of ROS and higher PMRS activity than their parent cells [137,138]. This suggests that induced expression of key enzymes in PM electron transport is coupled to glycolysis in the mitochondria-deficient cells. This relationship allows these cells to survive without causing further ROS production [109,139]. Interestingly, overexpressed PM redox enzymes, such as NADH-quinone oxidoreductase 1 (NQO1) and cytochrome b5 reductase (b5R), induce enhanced mitochondrial functions, such as increased activity of complexes I and III, and elevated ATP generation, without causing further ROS production [140,141]. These findings suggest that activating the PM redox enzymes increases the higher NAD(P)^+^/NAD(P)H ratio and enables more efficient mitochondrial functioning. Ultimately, these features enable cells to maintain redox homeostasis and energy metabolism under conditions of metabolic and energetic stress.

## 7. NQO1 as a Potential Target for Cancer Therapy

The enzymes NQO1 and b5R might be good therapeutic targets for cancer and age-related diseases. b5R overexpressing transgenic mice had enhanced mitochondrial functions and attenuated oxidative damage, as well as a modest extension in their life-spans compared with normal mice. These transgenic mice also exhibited a lower incidence of liver cancer after exposure to diethylnitrosamine and lower inflammatory markers than normal mice [142]. In addition, b5R inhibits angiogenesis and attenuates tumor formation in nasopharyngeal carcinoma [143]. Finally, b5R can detoxify arylhydroxylamine carcinogens [144].

Although NQO1 is highly expressed in some cancers, including pancreatic, lung, and breast cancers [145,146,147], other cancer cells have lower NQO1 activity than normal cells [145,148]. Therefore, there is a need for two different approaches to treat cancer cells with different NQO1 levels. Some antitumor agents become activated by NQO1 and then attack cancer cells. Mitomycin C, which is isolated from *Streptomyces caespitosus*, can form inter-strand DNA crosslinking under hypoxic and acidic conditions [146,149]. NQO1-activated β-lactone generates O_2_^•−^ and H_2_O_2_ in association with reduced pyridine nucleotides, resulting in autophagy-mediated cell death [150]. β-lactone has been used in Phase I and II clinical trials for cancer therapy because it inhibits topoisomerase I and then blocks the DNA repair system in mammalian cells [149]. NQO1 can also convert quinone derivatives, such as geldanamycin and 17-AAG, to hydroquinone forms. Geldanamycin has multiple effects such as the inhibition of DNA and RNA replication, blocking vSrc activity and attenuation of the Hsp90-mediated maturation of oncogenic proteins (e.g., HER2 and Raf-1) and steroid hormone receptors [149]. Therefore, upregulation of NQO1 can be combined with new NQO1-activated antitumor agents and cause preferential damage to cancer cells with high NQO1 activity [151,152]. For instance, ionization therapy at 2 Gy induces NQO1 upregulation and makes cells more vulnerable to β-lactone [153,154].

Different approaches are required for cancer cells with lower NQO1 activity than normal cells. Based on loss-of-function disease mechanisms, intact NQO1 might be required for normal cells to survive under stressful conditions because polymorphic forms of NQO1 are associated with an increased risk of cancer and neurodegenerative diseases (such as AD) [155]. Polymorphisms of NQO1 (e.g., C609T) are also closely related to a high incidence of various cancers [156,157,158]. Furthermore, mutated NQO1*2 is degraded rapidly by the 26S proteasome [159], and a deficiency in NQO1 makes cells more susceptible to benzene, which is a risk factor for hematological malignancy [160,161]. There is no significant interaction between p53 and the C609T polymorphism of NQO1 [162,163] because only the wild type NQO1 is normally involved in the stabilization of tumor suppressor protein p53 [164,165].

Moreover, NQO1 is responsible for a 2-electron transfer without forming semi-quinone radicals [163,166]. It is induced through the Nrf2-Keap1 pathway [167] (Figure 4). Under normal conditions, Nrf2 is tightly bound to Keap1, and is degraded in the 26S proteasome. However, oxidative stress or an insult that disrupts the disulfide bond can lead to the release of free Nrf2, which can translocate into the nucleus where it complexes with c-Jun. The Nrf2-c-Jun complex acts as the transcription factor for ARE and expresses a series of detoxifying enzymes, including NQO1 and HO-1. Previous research has shown that phytochemicals that target Nrf2 (e.g., sulforaphane and curcumin) can be very effective in cancer treatment. Sulforaphane, which is known to have chemopreventive activity on its own [168,169], has a synergic effect with the antitumor agent cisplatin [170]. Together, sulforaphane and cisplatin can be used with other phytochemicals (e.g., piperine and thymoquinone) to treat breast cancer [171]. Curcumin also exhibits many pharmacological effects, including anti-inflammatory, antioxidant, and anticancer effects [172,173]. These previous studies suggest that inducing detoxifying enzymes (including NQO1) through the Nrf2-Keap1 pathway can improve energy metabolism and be used to prevent cancer progression.

## 8. Conclusions

Cancer cells acquire nutrients and other factors from neighboring normal cells to proliferate, invade tissues, and metastasize. Signaling molecules produced by CAFs in the CME allow cancer cells to induce proliferation and metastasis through angiogenesis, immune tolerance, and ECM remodeling. Cancer cells can adapt to new environmental restrictions by changing their signaling pathways and metabolic dependence. Energy metabolism in cancer cells is very different from that in normal cells. Research on cancer bioenergetics has shown that metabolic reprogramming from mitochondrial respiration to glycolysis is associated with increased lactate fermentation during tumorigenesis and proliferation. Previous studies of cancer cell bioenergetics showed that these cells stimulate glycolysis and lactate fermentation to enhance glucose uptake and inhibit oxidative phosphorylation. The Warburg effect is an adaptive process in cancer cells that is related to angiogenesis, hypoxia and protection from excess ROS production. Although some cancers still rely on mitochondrial respiration for energy metabolism, the inhibition of glycolysis can be a potential therapeutic target in tumors that depend on glycolytic ATP production. These tumors inhibit oxidative phosphorylation to keep ROS levels below their threshold [174]. However, glycolysis inhibition is not effective in cancers with high mitochondrial functions. Inducing high levels of ROS through the use of ROS inducers or energy restriction could inhibit the proliferation of cancer cells.

Therefore, one key for novel cancer therapy is redox homeostasis and a metabolic shift toward enhanced mitochondrial functions. ROS production beyond the threshold of cancer cells is one therapeutic target, because their modest ROS levels are higher than normal cells. A pro-oxidant therapy using ROS inducers alone or in combination can be used to specifically damage cancer cells with high NQO1 levels. Alternatively, in cancer cells with low NQO1 activity, an antioxidant approach is required to keep normal cells intact during long-term chemotherapy or radiotherapy. Stabilizing a tumor suppressor, p53, and improving mitochondrial functions through NQO1 upregulation in association with the Nrf2-Keap1 pathway are also good therapeutic approaches to cancer treatment [175,176,177]. Along with low doses of anticancer agents, approaches that use activators of Nrf2 or NQO1, such as phytochemicals, can effectively reduce cancer proliferation and metastasis without causing concomitant cytotoxic effects on normal cells. Taken together, these potential therapeutic strategies for cancer prevention and treatment include inducing ROS production in cancer cells, and activating cell survival signaling and promoting mitochondrial functions in normal cells.

## Figures and Tables

**Figure 1 cancers-12-01822-f001:**
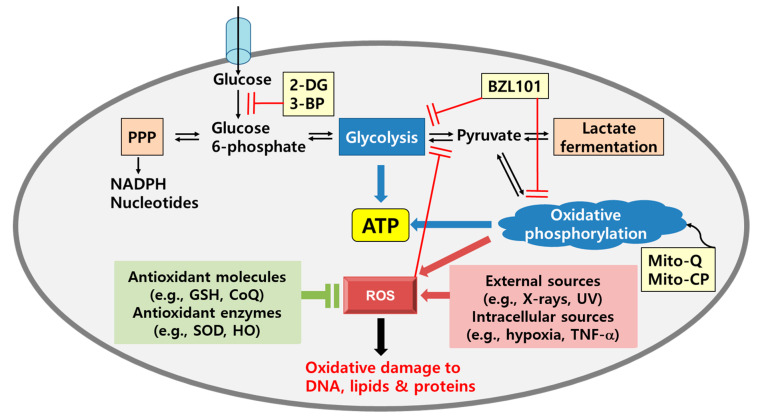
Metabolic fates of glucose metabolism. Under aerobic conditions, glucose is converted to pyruvate and then moved to the mitochondria, where it undergoes oxidative phosphorylation for ATP production. The mitochondria are the main source of ROS, which cause oxidative damage to biomolecules. O_2_^•−^ and its product H_2_O_2_ can be neutralized by antioxidant molecules (e.g., GSH and coenzyme Q) and antioxidant enzymes (e.g., SOD and catalase). Under anaerobic conditions (including in cancer cells), pyruvate is used for lactate fermentation to produce ATP. Glucose 6-phosphate is bypassed to the pentose phosphate pathway to generate NADPH. Oxidative stress and metabolic inhibitors are involved in metabolic reprogramming in cancer cells. Abbreviations: 2-DG, 2-deoxyglucose; 3-BP, 3-bromopyruvate; BZL101, Bezielle; Mito-Q, mitoquinone mesylate; PPP, pentose phosphate pathway; ROS, reactive oxygen species.

**Figure 2 cancers-12-01822-f002:**
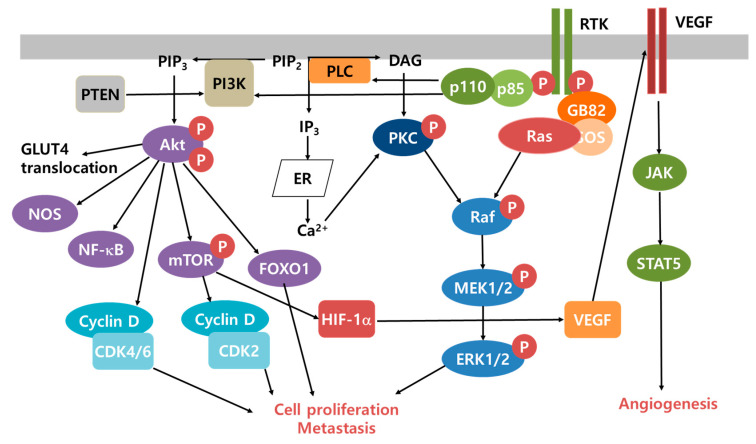
Proliferation, metastasis, and angiogenesis in cancer cells through the Akt, ERK, and STAT signaling cascades. PIP_2_ is converted to PIP_3_ by PI3K, which activates Akt signaling. Alternatively, PIP_2_ can be degraded by PLC into DAG and IP_3_. Remaining in the PM, DAG activates PKC. Next, IP_3_ binds to the IP_3_ receptor in the ER and triggers Ca^2+^ release from the ER, stimulating Ca^2+^-dependent PKC, which activates Raf, MEK, and ERK signaling cascades. After binding with the ligands, RTKs are autophosphorylated and recruit the GRB2 and SOS complex to change inactive Ras-GDP into active Ras-GTP. Activated Ras recruits p110 (a catalytic subunit of PI3K), and phosphorylates PIP_2_, which initiates survival signaling via Akt. Autophosphorylated RTKs can bind to p85 (a regulatory subunit of PI3K), which then attaches to p110 and forms the active complex. mTOR-activated HIF-1a stimulates VEGF, which binds to VEGF ligand and triggers JAK-STAT5 signaling for angiogenesis. Abbreviations: Akt, protein kinase B; CDK, cyclin-dependent kinase; DAG, diacylglycerol; FOXO1, Forkhead family of transcription factor 1; GRB2, growth factor receptor-bound protein 2; HIF-1α, hypoxia-inducing factor 1α; JAK, Janus kinase; MEK1/2, mitogen-activated protein kinase kinase 1 and 2; mTOR, mammalian target of rapamycin; p100, phosphor- NF-kB2; p85, regulatory subunit of PI3K; PKC, protein kinase C; PIP_2_, phosphatidylinositol 4,5-bisphosphate; PIP_3_, phosphatidylinositol 3,4,5-trisphosphate; PLC, phospholipase C; PTEN, phosphatase and tensin homolog; Raf, rapidly accelerated fibrosarcoma; Ras; SOS, salt overly sensitive; STAT5, signal transducer and activator of transcription 5.

**Figure 3 cancers-12-01822-f003:**
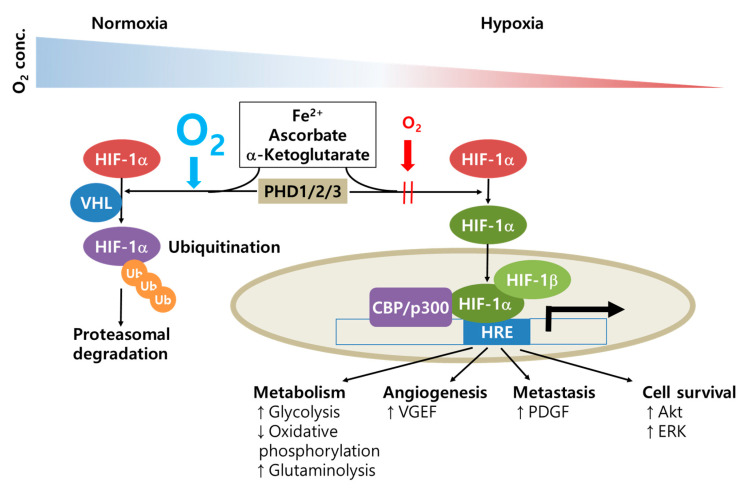
Oxygen-dependent regulation of HIF-1α in cancer cells. Under normoxia, HIF-1α is recognized and hydroxylated by PHD in the presence of Fe^2+^, ascorbate, and α-ketoglutarate. Hydroxylated proline in HIF-1α is recognized by VHL and subsequently ubiquitinated for proteasomal degradation. However, under hypoxic conditions, HIF-1α is stabilized and translocated into the nucleus, where it is dimerized with HIF-1β. Following the binding of the HIF-1α-HIF-1β complex with other cofactors (such as CBP/p300), the molecules can target the HRE, which is involved in the metabolic switch, angiogenesis, metastasis, and cell survival. Abbreviations: CBP, cAMP response element binding protein; HRE, hypoxia response elements; p300, E1A binding protein; PHD, proxyl hydroxylase protein; Ub, ubiquitin; VHL, von Hippel–Lindau protein.

**Figure 4 cancers-12-01822-f004:**
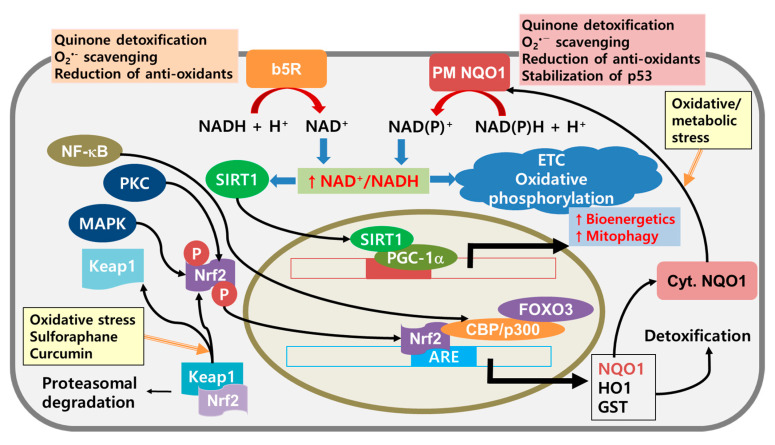
Redox homeostasis and metabolic shift through activation of the Nrf2-Keap1 pathway and regulation of the NAD^+^/NADH ratio by PM redox enzymes. Disulfide bonds between Nrf2 and Keap1 are broken by oxidative stress and other Nrf2 activators. Activated Nrf2 is translocated into the nucleus, combined with CBP/p300 and FOXO3, and bound to ARE, which results in the expression of detoxifying enzymes. Under further oxidative/metabolic stress, expressed cytosolic NQO1 can be translocated into the inner surface of the PM. A high NAD+/NADH ratio is induced by stimulating b5R and PM NQO1 to activate SIRT1-PGC-1α signaling and enhance mitochondrial bioenergetics. Abbreviations: ARE, antioxidant response element; CBP, cAMP response element binding protein; Cyt. NQO1, cytosolic form of NADH-quinone oxidoreductase 1; FOXO3, Forkhead family of transcription factor 3; GST, glutathione S-transferase; p300, E1A binding protein; p53, tumor suppressor encoded by TP53; PKC, protein kinase C; PM NQO1, plasma membrane bound NQO1; PGC-1α, peroxisome proliferator-activated receptor-γ coactivator).

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
