# Peer review of "Insights into the New Cancer Therapy through Redox Homeostasis and Metabolic Shifts"

_cancers, 2020, doi:10.3390/cancers12071822_

Round 1

Reviewer 1 Report

The manuscript needs revision, as indicated below:

  • the abbreviations should be described in the first time they are used in the text (in the Abstract and in the first time it has been used in the text);
  • Introduction, lines 41-42: this part of the text is confusing. It is suggested describing that the electron transfer chain (ETC) components include the complexes I-IV and that the OXPHOS system involves the ETC components and the complex V;
  • Introduction, lines 42-44: this part of the text should be better described, since it is not clear the link between this part of the paragraph and the lines 41-42;
  • section 2, lines 54-55: "several compounds"? This paragraph also lacks references;
  • section 2, lines 72-74: this part of the text lacks references. This should be revised throughout the text;
  • the utilization of abbreviatures should be revised in the whole text (mainly the order of utilization of the abbreviatures);
  • section 2, lines 84-95: this part of the text is confusing and should be rewritten. It is suggested describing the antioxidant agents as follows: endogenous (non-enzymatic and enzymatic) and exogenous (mainly non-enzymatic). It is suggested because enzymes are mainly endogenous and it is not necessary describing it as a different class. It can be misunderstood by the Readers of the Journal. Also, the following phrase is not adequate: "Anti-oxidant molecules contain...". Suggestion: "Glutathione (GSH), alpha-lipoate,..., can be cited as non-enzymatic endogenous antioxidant agents". This is just a suggestion, but the major classification of antioxidant (endogenous versus exogenous) should be done as described above;
  • the imbalance between oxidants and antioxidants may result also in necrosis, and not only apoptotis (please, see lines 90-92);
  • section 2, lines 107-109: is oxidative stress the cause of those diseases? Or is oxidative stress a consequence of those diseases? It is suggested describing oxidative stress as a part of the pathophysiological alterations seen in those diseases;
  • section 4, lines 178-179: "Pyruvate, which is GENERATED from glucose during glycolysis";
  • section 4, lines 182-183: this part should be revised and rewritten;
  • pzero cells: lines 310-312 versus lines 242;
  • typo errors should be checked and corrected ("curcumine", others);
  • the figures are excellent;
  • the text is not speculative.    

Author Response

Dear, Reviewer,

Enclosed is my point-by-point response to your comments. I am grateful to the reviewers for your helpful comments, which have allowed me to improve the manuscript considerably. 

Best regards

Dong-Hoon Hyun, PhD

Reviewer 2 Report

The proposed review manuscript by Hyun focuses on the interaction between reactive oxygen species (ROS) and energy metabolism in cancer cells in terms of the development of novel strategies for cancer therapy.

This is a very intriguing are very well written review that follows a clear logical flow. As a general suggestion, the author may consider to pay more attention to the cancer microenvironment, since the whole process of cancer initiation, growth and spread to distant tissues is based on a tight interaction between the cancer cells and cancer microenvironment.

Specific minor comment:

  • Chapter 2, lines 56-57: Please provide a reference.

Author Response

Dear, Reviewer,

Enclosed is my point-by-point response to your comments. I am grateful to the reviewers for your helpful comments, which have allowed me to improve the manuscript considerably. 

Best regards,

Dong-Hoon Hyun, PhD

Reviewer 3 Report

Thanks to the authors for the purpose. Unfortunately, I find the paper not suitable for the journal. To me, this paper can find a place in a biology or cancer-biology dedicated journal. 

The review is prolix and not well focused on the topic. Discussion is poor and the review lacks of adquate conclusions.

I would have focused only on redox homeostasis going deeper into this topic.

Author Response

(The authors gave the same response as above.)

Round 2

Reviewer 3 Report

The authors improved significantly their paper, now suitable for publication